# Investigations of the Influence of Two Pyridyl-Mesoionic Carbene Constitutional Isomers on the Electrochemical and Spectroelectrochemical Properties of Group 6 Metal Carbonyl Complexes

**Tobias Bens** [1,2] **and Biprajit Sarkar** [1,2,*]

1   Institut für Anorganische Chemie, Universität Stuttgart, Pfaffenwaldring 55, 70569 Stuttgart, Germany; tobias.bens@uibk.ac.at
2   Institut für Chemie und Biochemie, Freie Universität Berlin, Fabeckstraße 34–36, 14195 Berlin, Germany
*   Correspondence: biprajit.sarkar@iac.uni-stuttgart.de

**Abstract:** Metal complexes of mesoionic carbenes (MICs) of the triazolylidene type and their derivatives have gained increasing attention in the fields of electrocatalysis and photochemistry. The redox activity of these metal complexes is critical for their applications in both the aforementioned fields. Easy accessibility and modular synthesis open a wide field for the design of ligands, such as bidentate ligands. The combination of an MIC with a pyridyl unit in a bidentate ligand setup increases the $\pi$ acceptor properties of the ligands while retaining their strong $\sigma$ donor properties. The analogy with the well-established 2,2′-bipyridine ligand allows conclusions to be drawn about the influence of the mesoionic carbene (MIC) moiety in tetracarbonyl group 6 complexes in cyclic voltammetry and (spectro)electrochemistry (SEC). However, the effects of the different connectivity in pyridyl-MIC ligands remain underexplored. Based on our previous studies, we present a thorough investigation of the influence of the two different pyridyl-MIC constitutional isomers on the electrochemical and the UV-vis-NIR/IR/EPR spectroelectrochemical properties of group 6 carbonyl complexes. Moreover, the presented complexes were investigated for the electrochemical conversion of $CO_2$ using two different working electrodes, providing a fundamental understanding of the influence of the electrode material in the precatalytic activation.

**Keywords:** mesoionic carbenes; (spectro)electrochemistry; carbonyl ligands; group 6 carbonyls; EPR spectroscopy

## 1. Introduction

In 2001, Sharpless and co-workers coined the term "click" chemistry to describe modular reactions with a wide scope and high yields, producing only mild inoffensive byproducts [1]. The azide–alkyne cycloaddition reaction is arguably one of the best examples of a click reaction. The thermally induced 1,3-dipolar cycloaddition between alkynes and azides results in a mixture of two regioisomers [2]. In 2002, two groups independently discovered the copper-catalyzed azide–alkyne cycloaddition reaction (CuAAC), generating exclusively the 1,4-regioisomer of 1,2,3-triazole [3,4].

The methylation of 1,2,3-triazoles leads to the formation of so-called triazolium salts in near quantitatively yields [5–7]. They represent one of the most important precursors for triazolylidenes, a class of carbenes that are better known as abnormal *N*-heterocyclic carbenes (aNHCs) or mesoionic carbenes (MICs). This classification arises from the fact that while following octet rules, no resonance structures can be drawn for MICs without charge separation, unlike their well-established *N*-heterocyclic carbene (NHC) counterparts [5,7–9]. Therefore, not surprisingly, the synthetic scope of MICs has expanded rapidly, opening up the possibility of introducing additional donor substituents, such as pyridine, to generate

bidentate ligands [10–16] or post-modifications to *N*-heterocyclic olefins (NHOs) [17,18] and mesoionic imines (MIIs) [19,20], which are promising candidates for small molecule activation [5].

Suntrup et al. showed in 2017 that the insertion of a pyridyl moiety into 1,2,3-triazole- and 1,4-triazolylidene-based Re(I) carbonyl complexes drastically improves the overall $\pi$ acceptor character of the ligand, while the incorporation of an MIC unit results in a greater $\sigma$ donor strength compared to the well-established bpy ligand [21]. The robustness toward reductive electrochemistry provided the basis for the investigation of a series of pyridyl-MIC Re(I) complexes in the electrochemical reduction in $CO_2$ to generate CO with high selectivity and the study of their photophysical properties [16].

However, many of the most promising electrocatalysts explored contain expensive and rare metals, which preclude their large-scale applications [22–27]. In recent years, great efforts have been made to develop more earth-abundant photo- and electrocatalysts for the activation of small molecules based on carbenes [28–39].

Group 6 metal complexes are attractive candidates because of their natural occurrence, such as molybdenum in the active site of enzymes that convert $CO_2$ to formate [40].

Recent reports have shown that the isoelectronic and isostructural group 6 metal complexes of [M(bpy)(CO)$_4$] (M = Cr, Mo, W) and [M(L)(CO)$_4$] (L = "non-innocent" ligands) with Mo and W are capable of electrocatalytic conversion of $CO_2$ [28,29,33,36,37,39].

Tory et al. and Clark et al. reported the (spectro)electrochemical properties of group 6 complexes [M(bpy-R)(CO)$_4$] (R = 5,5′ H, 5,5′ $^t$Bu) and demonstrated their activity in $CO_2$ reduction on a gold (Au WE) and glassy carbon working electrode (GC WE), respectively [28,39]. The results indicate two important facts: first, the substitution of the bpy moiety results in a shift of the reduction potential for the precatalytic activation, and second, the change in the working electrode from a platinum working electrode (Pt WE) to a Au WE shifts the onset potential for electrocatalytic $CO_2$ reduction by +0.6 V, similar to what was reported for the group 7 electrocatalysts [25]. Based on these results, Neri et al. investigated the role of the electrode–catalyst interaction using vibrational sum frequency generation spectroscopy (VSFG), providing an insight into the mechanism at the electrode surface [36]. Cyclic voltammetric measurements with a Au WE show an equilibrium between the one-electron reduced species [Mo(bpy)(CO)$_4$]$^-$ and [Mo(bpy)(CO)$_3$]$^-$ after CO dissociation. In contrast, using a Pt WE, two-electron reduction is required to generate the precatalytically active species.

Recently, we have presented a series of two 1,4-pyidyl-MIC group 6 carbonyl complexes [M(**L**)(CO)$_4$] (M = Cr, Mo, W) with two different constitutional isomers (**L**: **C–C** = pyridyl-4-triazolylidene [41] and **C–N** = pyridyl-1-triazolylidene [42–44]) that exhibit excellent photophysical and photochemical properties, making them suitable candidates in photo-induced small molecule activation [43–46]. For the first time, details of the influence of the two constitutional isomers were reported in the chemically and electrochemically oxidized [Cr(**L**)CO)$_4$]$^+$ complexes, providing detailed insights into the extraordinary $\sigma$ donor properties [41]. In addition, a comprehensive study of precatalytic activation in [Rh(Cp*)] complexes for electrochemical H$^+$ reduction was reported, demonstrating the capability of small molecule activation with both ligands (Scheme 1) [47].

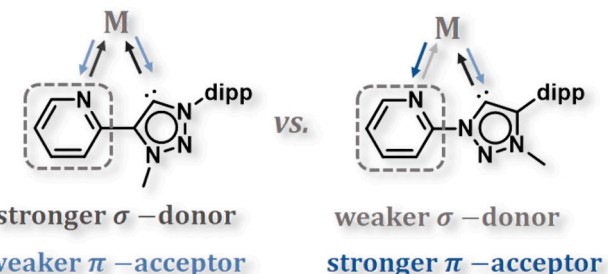

**Scheme 1.** Relative donor/acceptor strength of constitution isomers: **C–C** (**left**) and **C–N** (**right**).

Based on our previous studies, we report a comprehensive electrochemical and spectroelectrochemical investigation of [M(**C–C**)(CO)$_4$] and [M(**C–N**)(CO)$_4$] [42] (M = Cr, Mo, W) to gain a fundamental understanding of the effects of the two constitutional isomers on their electronic structures and perform reactivity of the complexes in electrochemical $CO_2$ reduction as a function of the electrode material.

## 2. Results and Discussion

The triazolium salts, [H(**C–C**)](BF$_4$) [21] and [H(**C–N**)](Otf) [42], and the complexes, [M(**C–C**)(CO)$_4$] [41] and [M(**C–N**)(CO)$_4$] [42,43], were synthesized according to a previously reported protocol (Scheme 2).

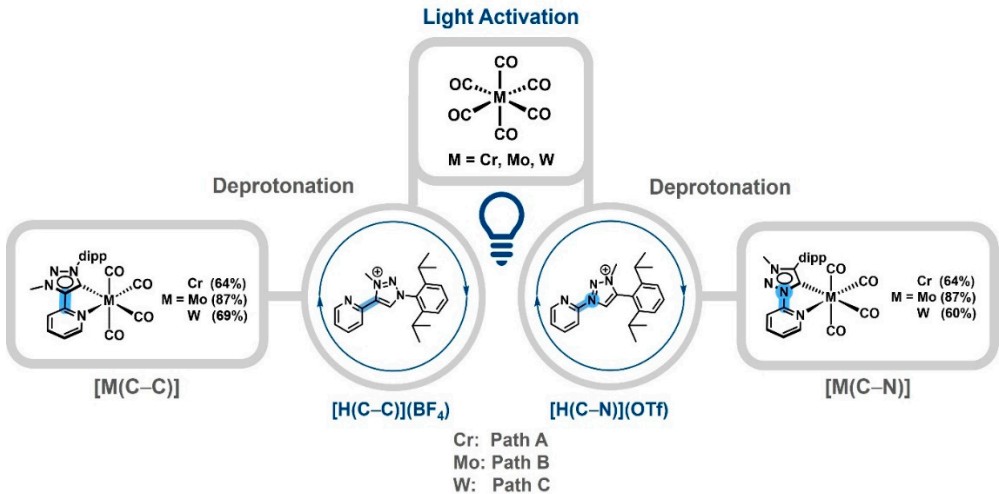

**Scheme 2.** Synthetic protocol for [M(**L**)CO)$_4$] (M = Cr, Mo, W; Path A [42], Path B [42], Path C [43]).

The light-induced activation of the corresponding [M(CO)$_6$] followed by the addition of [H(**C–C**)](BF$_3$) or [H(**C–N**)](OTf) and subsequent deprotonation with NEt$_3$ leads to the chromium and tungsten complexes, [M(**C–C**)(CO)$_4$] and [M(**C–N**)(CO)$_4$], after chromatographic workup and recrystallization, while in the case of molybdenum, the precursor [Mo(nbd)(CO)$_4$] (nbd = norbornadiene) was synthesized and further converted in the presence of a base to isolate [Mo(**C–C**)(CO)$_4$] or [Mo(**C–N**)(CO)$_4$], respectively.

### 2.1. Cyclic Voltammetry with a GC WE and EPR-SEC

The redox potentials measured from cyclic voltammetry are often, but not always, used for gauging the donor/acceptor properties of the ligands in metal complexes. A reversible metal-centered oxidation, as observed for [Cr(**C–C**)(CO)$_4$] [41] and [Cr(**C–N**)(CO)$_4$], [42] allows us to estimate the overall $\sigma$ donor strength of the ligand, while a reversible ligand-centered reduction can be used to determine indirectly the $\pi$ acceptor capacity of the ligand.

Previous reports from our group already established a stronger $\sigma$ donor strength of the ligand in [Cr(**C–C**)(CO)$_4$] compared to [Cr(**C–N**)(CO)$_4$] [41,42]. The same trend in this regard is observed for the higher homologs, [M(**C–C**)(CO)$_4$] and [M(**C–N**)(CO)$_4$] (M = Mo, W). However, the oxidations of the respective complexes are irreversible as a consequence of the kinetic lability of the CO ligands and the possibility of forming complexes with higher coordination numbers in the oxidized complexes (Figure 1; see Supplementary Materials S6) [48,49]. The oxidation potentials follow the trend according to the ionization energy of the central metal atom (Cr > Mo > W) [50].

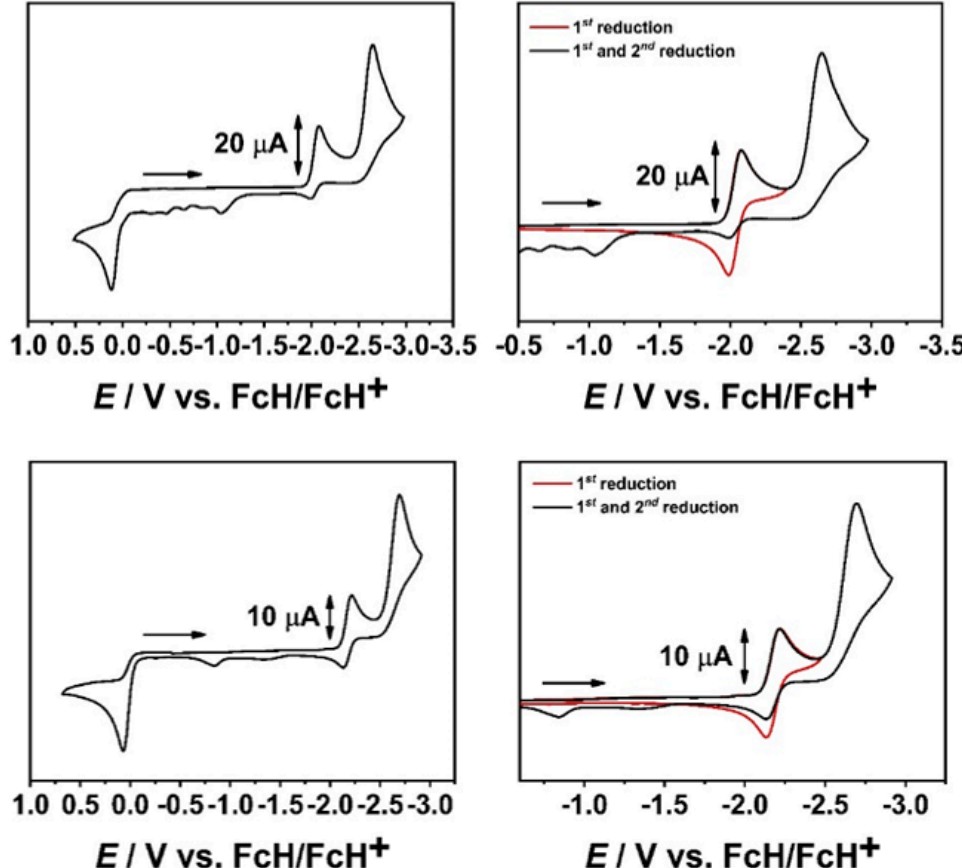

**Figure 1.** Cyclic voltammograms of [W(**C–C**)(CO)$_4$] (**top**) and [W(**C–N**)(CO)$_4$] (**bottom**) in CH$_3$CN and 0.1 M Bu$_4$NPF$_6$ at a scan rate of 100 mV/s and a glassy carbon working electrode.

All presented complexes, on the other hand, show a reversible first reduction, followed by a second irreversible reduction, whereas in the series of [M(**C–N**)(CO)$_4$], a third reduction process is observed at lower scan rates (Table 1; see Supplementary Materials S2) [42].

**Table 1.** Redox potentials of [M(**C–C**)(CO)$_4$] and [M(**C–N**)(CO)$_4$] (M = Cr, Mo, W) in CH$_3$CN and 0.1 M NBu$_4$PF$_6$ at 100 mV/s vs. FcH/FcH$^+$ (FcH = ferrocene) with a glassy carbon working electrode.

| | 1. Red./V | | 2. Red./V | 1. Ox./V | |
| --- | --- | --- | --- | --- | --- |
| | $E_{1/2}^{red1}$ | $\Delta E_p$ | $E_p^{red2}$ | $E_{1/2}^{ox1}$ | $\Delta E_p$ |
| [Cr(**C–C**)(CO)$_4$] [41] | −2.26 | 0.07 | −2.80 | −0.21 | 0.07 |
| [Cr(**C–N**)(CO)$_4$] [42] | −2.16 | 0.07 | −2.79 | −0.17 | 0.07 |
| [Mo(**C–C**)(CO)$_4$] | −2.21 | 0.07 | −2.70 | 0.07 $^a$ | |
| [Mo(**C–N**)(CO)$_4$] [42] | −2.10 | 0.08 | −2.68 | 0.08 $^a$ | |
| [W(**C–C**)(CO)$_4$] | −2.19 | 0.08 | −2.69 | 0.07 $^a$ | |
| [W(**C–N**)(CO)$_4$] | −2.05 | 0.06 | −2.65 | 0.12 $^a$ | |

$^a = E_p^{ox1}$.

The reduction potentials $E_{1/2}^{red1}$ presented in Table 1 are in good agreement with the aforementioned $\pi$ acceptor properties of the constitutional isomers. In the case of [M(**C–N**)(CO)$_4$] (M = Cr, Mo, W), the first reduction is shifted to more anodic potential compared to [M(**C–C**)(CO)$_4$] (M = Cr, Mo, W), indicating the greater $\pi$ acceptor ability of the **C–N** linked constitutional isomer in the complexes.

To obtain detailed insights into the electronic structure of the first reduction, electron paramagnetic resonance (spectro)electrochemistry (EPR-SEC) was performed with a Au WE in 0.1 M Bu$_4$NPF$_6$/CH$_3$CN (Figure 2 and Table 2; see Supplementary Materials S3).

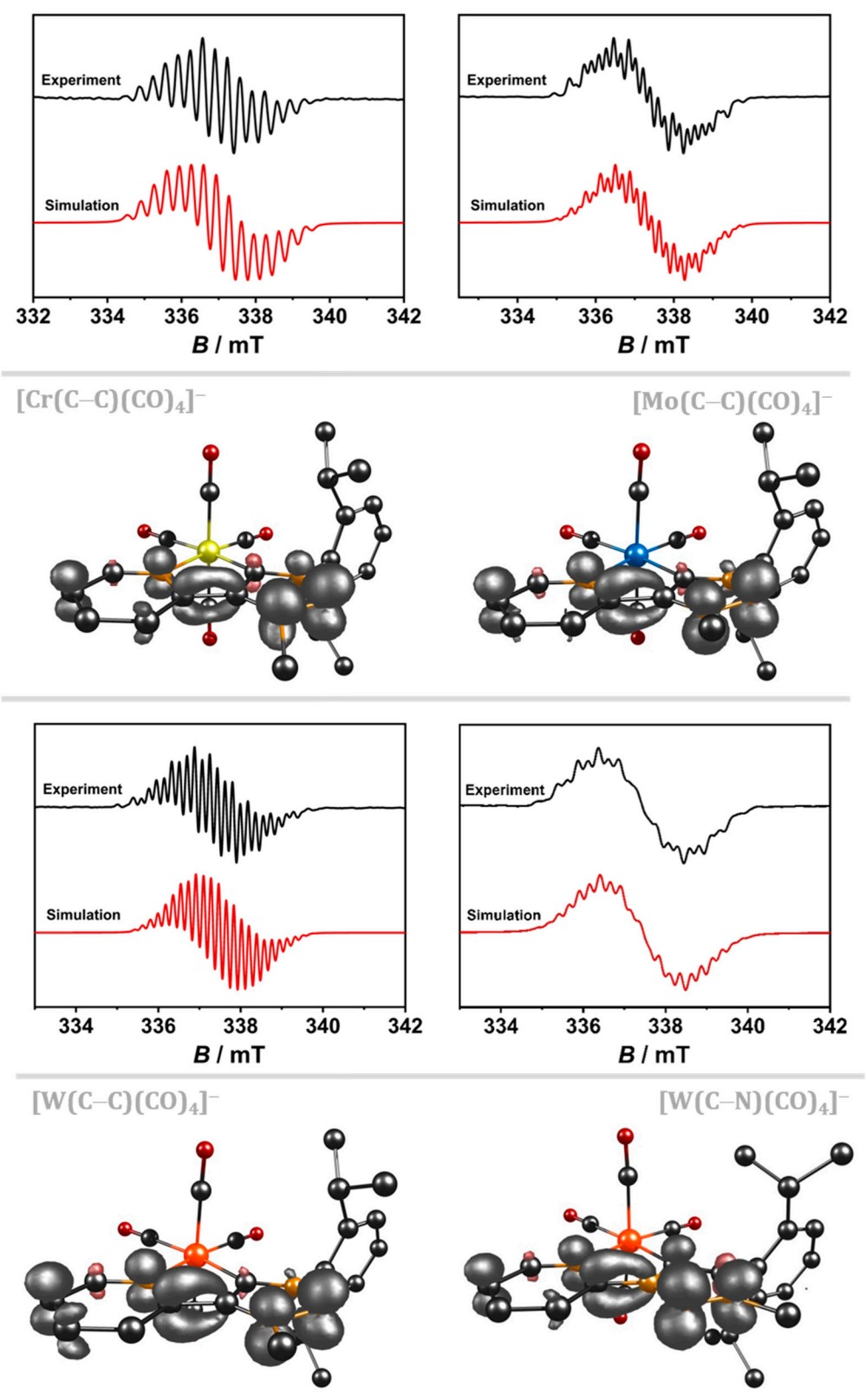

**Figure 2.** EPR spectrum and spin density plot (B3LYP/RIJCOSX/D3BJ/def2-TZVP, iso value = 0.006) of [Cr(**C–C**)] (**top left**), [Mo(**C–C**)] (**top right**), [W(**C–C**)] (**bottom left**), and [W(**C–N**)] (**bottom right**) in 0.1 M $NBu_4PF_6$/$CH_3CN$ with a Au working electrode during the first reduction (black: experimental, red: simulation).

**Table 2.** EPR simulation data of $[M(\mathbf{C–N})(CO)_4]^-$ (M = Cr, Mo, W) and $[W(\mathbf{C–N})(CO)_4]^-$.

| | $[Cr(\mathbf{C–C})(CO)_4]^-$ | $[Mo(\mathbf{C–C})(CO)_4]^-$ | $[W(\mathbf{C–C})(CO)_4]^-$ | $[W(\mathbf{C–N})(CO)_4]^-$ |
|---|---|---|---|---|
| $g$ | 2.0030 | 2.0033 | 2.0028 | 2.0032 |
| AM | 5.90 | 1.97 | 13.10 | 2.65 |
| AN1 | 17.90 | 10.37 | 10.39 | 16.92 |
| AN2 | 17.80 | 4.95 | 5.60 | 14.22 |
| AN3 | 11.00 | 16.10 | 5.60 | 6.47 |
| AN4 | 9.60 | 9.60 | 4.53 | 5.82 |
| AH1 | 11.70 | 12.00 | 4.16 | 16.20 |
| AH2 | 9.50 | 11.90 | 20.27 | 13.77 |
| AH3 | 3.00 | 10.50 | 20.27 | 13.22 |
| AH4 | 2.00 | 14.45 | 15.36 | 7.19 |
| AH5 | - | - | 1.01 | 3.19 |
| AH6 | - | - | 1.01 | 3.19 |
| AH7 | - | - | 0.63 | - |
| AH8 | - | - | 1.81 | - |
| AH9 | - | - | 0.49 | - |
| AH10 | - | - | 0.50 | - |
| lwpp $^a$/mT | [0 0.123] | [0 0.121] | [0 0.054] | [0 0.161] |

$^a$ The first value corresponds to the Gaussian and the second to the Lorentzian shape.

Upon a reduction in room temperature, line-rich EPR spectra at $g$ = 2.003 are observed for all complexes, showing hyperfine coupling to all four $^{14}$N nuclei within the central pyridyl-MIC ligand framework (Table 2). The hyperfine coupling constants of the $^{14}$N nuclei and the spin density plots of the respective complexes reveal a strong interaction of the electron spin with the N$^2$ and N$^3$ nuclei of the reduced 1,2,3-triazolylidene (MIC) moiety, and to a smaller extent, with the $^{14}$N nuclei of the pyridyl-*N* and the N$^1$ nuclei of the MIC unit. Only in the case of $[W(\mathbf{C–C})]^-$ is a strong coupling to only one $^{14}$N nucleus observed. A plausible explanation might be the stronger delocalization of the electron spin within the **C–C** isomer. The $[W(\mathbf{C–C})]^-$ complex shows $^1$H hyperfine coupling to ten $^1$H nuclei. In contrast, the analog $[Cr(\mathbf{C–C})(CO)_4]^-$ and $[Mo(\mathbf{C–C})(CO)_4]^-$ complexes display $^1$H hyperfine coupling to four $^1$H nuclei, which can be assigned to the pyridyl-*H*. The strong interaction of the electron spin within the pyridyl moiety is also present in $[W(\mathbf{C–C})(CO)_4]^-$. The complex shows a strong $^1$H coupling constant to four $^1$H nuclei, indicating a predominant localization within the central pyridyl-MIC framework. However, small hyperfine couplings with six additional $^1$H nuclei are observed. Even though the spin density plot of $[W(\mathbf{C–C})]^-$ does not directly indicate the localization of the electron spin at the different ligand fragments, the coupling to three $^1$H nuclei of the methyl group at the MIC moiety and three $^1$H nuclei of the 2,6-diisopropylphenyl (=dipp) substituent are reasonable.

The influence of the constitutional isomers is shown in the EPR spectrum of $[W(\mathbf{C–C})(CO)_4]^-$ and $[W(\mathbf{C–N})(CO)_4]^-$. The analog tungsten C–N complex displays a stronger coupling of the electronic spin with the four $^{14}$N nuclei in the pyridyl-MIC moiety. Consequently, the line-rich spectrum shows an increased line broadening of the isotropic signal. In contrast to its C–C counterpart, only six $^1$H hyperfine couplings are observed in $[W(\mathbf{C–N})(CO)_4]^-$. This observation could indicate an increased localization of the electron spin at the MIC moiety, consequently leading to a decreased contribution of the dipp substituent. The stronger localization at the MIC moiety in the C–N linked isomer is further affirmed by the stronger $^1$H hyperfine coupling to the methyl group. The EPR spectrum shows $^1$H coupling constants of up to 7.19 MHz, while the **C–C** linked analog shows only weak couplings of up to 1.81 MHz. Additionally, three strong $^1$H hyperfine couplings to three pyridyl-*H* are observed, confirming the significant localization at the pyridyl-MIC framework within the **C–N** isomer.

Unfortunately, no clear trend regarding the influence of the central metal atom in the series of [M(**C–C**)(CO)$_4$]$^-$ and [M(**C–N**)(CO)$_4$]$^-$ (M = Cr, Mo, W) could be observed, despite all metal ions showing a coupling with the ligand-centered radical [42].

To further shine a light on the influence of the constitutional isomers in [M(**C–C**)(CO)$_4$] and [M(**C–N**)(CO)$_4$] (M = Cr, Mo, W), IR-SEC with a Au WE in 0.1 M NBu$_4$PF$_6$/CH$_3$CN was conducted.

### 2.2. IR-Spectroelectrochemistry

In contrast to cyclic voltammetry, IR spectroscopy of the complexes, [M(**C–C**)(CO)$_4$] and [M(**C–N**)(CO)$_4$] (M = Cr, Mo, W), under investigation is a common method for the characterization of the electronic structure due to the characteristic CO stretching frequencies.

The IR spectra of [M(**C–C**)(CO)$_4$] and [M(**C–N**)(CO)$_4$] in CH$_2$Cl$_2$ show four CO stretching frequencies as a consequence of the lowering of symmetry around the metal center (Table 3, Figure 3). Even though the number of bands observed in the IR spectra are identical, their positions shifted, depending on the electronic nature of the ligands and the central metal atoms.

**Table 3.** CO stretching frequencies of [M(**C–C**)] and [M(**C–N**)] (M = Cr, Mo, W) in CH$_2$Cl$_2$.

| | $\tilde{\upsilon}$ (CO)/cm$^{-1}$ | | | | $\tilde{\upsilon}_{average}$ (CO)/cm$^{-1}$ |
|---|---|---|---|---|---|
| [Cr(**C–C**)(CO)$_4$] [41] | 1998 | 1890 | 1875 (sh) | 1822 | 1896 |
| [Cr(**C–N**)(CO)$_4$] [42] | 1998 | 1890 | 1878 (sh) | 1830 | 1899 |
| [Mo(**C–C**)(CO)$_4$] | 2004 | 1894 | 1876 (sh) | 1827 | 1900 |
| [Mo(**C–N**)(CO)$_4$] [42] | 2006 | 1896 | 1876 (sh) | 1830 | 1902 |
| [W(**C–C**)(CO)$_4$] | 1998 | 1882 | 1870 (sh) | 1826 | 1894 |
| [W(**C–N**)(CO)$_4$] | 2000 | 1884 | 1873 (sh) | 1830 | 1897 |

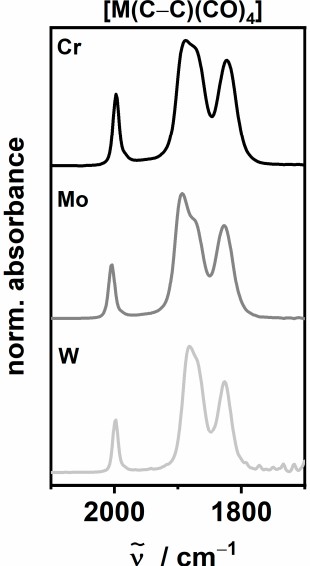 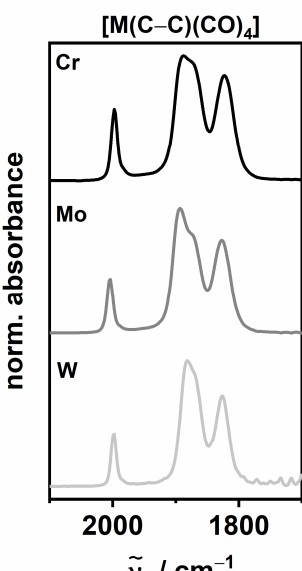

**Figure 3.** IR spectra of [M(**C–C**)(CO)$_4$] [41] (**left**) and [M(**C–N**)(CO)$_4$] [42] (**right**) in CH$_2$Cl$_2$ (M = Cr: black, Mo: grey, W: light grey).

Concerning the net electron density of the [M(CO)$_4$] fragment with the incorporated pyridyl-MIC ligands, the averaged CO stretching frequencies presented in Table 3 further confirm the greater $\sigma$ donor strength of the chelating ligand observed in [M(**C–C**)(CO)$_4$] compared to [M(**C–N**)(CO)$_4$].

The influence of the constitutional isomers becomes evident upon a one-electron reduction in [M(**C–C**)(CO)$_4$] and [M(**C–N**)(CO)$_4$] during IR-SEC (Figure 4; see Supplementary Materials S4). Within the series of [M(**C–N**)(CO)$_4$], the observation of isosbestic points during the IR-SEC measurements is consistent with a clean conversion of the native species into

the reduced $[M(C–N)(CO)_4]^-$ complexes. The shift of the frequencies by about 20 cm$^{-1}$ to lower wavenumbers confirms the predominantly ligand-centered reduction and is in good agreement with our calculated changes in the CO stretching frequencies of $[M(C–N)(CO)_4]^-$ (see Supplementary Materials S4) [41,42].

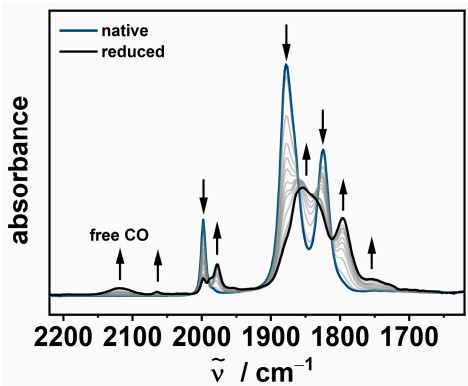 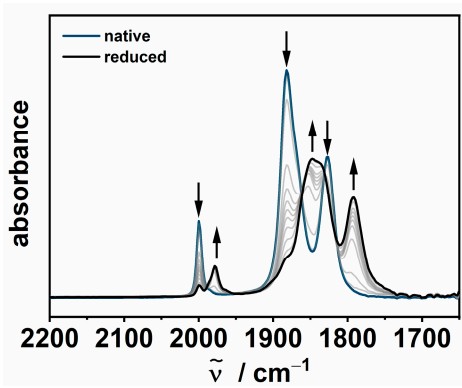

**Figure 4.** Changes in the IR spectra of $[W(C–C)(CO)_4]$ (**left**) and $[W(C–N)(CO)_4]$ (**right**) in CH$_3$CN/0.1 M Bu$_4$NPF$_6$ with a Au working electrode during the first reduction.

However, the picture changes upon a reduction in the other isomer. All complexes within the series show at least two new species in the IR-SEC measurements, as indicated by the formation of several new IR bands.

The most significant change can be assigned to the newly formed band at 2119 cm$^{-1}$. Torey et al. described a similar observation after a reduction in $[Mo(bpy)(CO)_4]$ [39]. The IR band at 2130 cm$^{-1}$ could be assigned to adsorbed CO at the Au electrode surface. Furthermore, in-depth investigations by VSFG by *Neri* et al. confirmed the dissociative EC mechanism of CO upon a reduction in a Au WE [36]. Based on these results and our theoretical calculations (see Supplementary Materials S4), the reduced species could likely be a mixture of the one-electron-reduced $[M(C–C)(CO)_4]^-$ species, the coordinatively unsaturated complex $[M(C–C)(CO)_3]^-$, and/or the solvent adduct $[M(C–C)(CH_3CN)(CO)_3]^-$, formed after subsequent CO dissociation.

In addition, a comparison of the IR spectra before and after electrolysis in the OTTLE cell clearly indicates the partial decomposition of $[M(C–C)(CO)_4]$ after reduction, whereas only minor decomposition products are observed in the series of $[M(C–N)(CO)_4]$ after prolonged electrolysis [42]. These results provide useful information on the stabilization of the ligand-centered radical based on the different linkage in the two constitutional isomers, as the CO cleavage observed in $[M(C–C)(CO)_4]$ gives access to an open coordination site for potential electrocatalytic applications, such as electrochemical CO$_2$ reduction [28,29,33,36,37,39].

To confirm the reversibility in the series of $[M(C–N)(CO)_4]$ and the EC mechanism observed for $[M(C–C)(CO)_4]$ upon reduction, UV/vis/NIR-SEC measurements were performed.

### 2.3. UV/vis/NIR-Spectroelectrochemistry

UV/vis/NIR-SEC is a commonly employed technique to test either pure electrochemical reversibility or reversibility following an EC mechanism [51].

All presented complexes display electronic transitions in the visible to near UV region (300–550 nm), which can be assigned to metal-to-ligand charge transfers (MLCTs) with an additional contribution of the axial CO ligands in the ground state and excited state (see Supplementary Materials S6.20–S6.40) [41,42]. Within the series of $[M(C–C)(CO)_4]$, the MLCT transitions are blue-shifted compared to $[M(C–N)(CO)_4]$, which is in good agreement with the previously described $\pi$ acceptor properties of the C–N linked constitutional isomer. However, a significant contribution of the aromatic substituent is observed in $[M(C–N)(CO)_4]$ (see Supplementary Materials S6.40) [42].

The electrochemical reduction in [M(**C**–**C**)(CO)$_4$] leads to broad transitions in the visible and NIR region (650–2100 nm; Figure 5 and Supplementary Materials S30 and S31). According to TD-DFT calculations, these bands can be assigned to an intra-ligand charge transfer (ILCT) from the reduced **C**–**C** linked ligand to the 2,6-diisopropylphenyl substituent and a ligand-to-ligand charge transfer (LLCT) from the reduced ligand to the axial CO ligands. The absorption bands in the 380–400 nm range are best described as metal ligand-to-ligand charge transfer (MLLCT) from the [M(CO)$_4$] fragment to the pyridyl-MIC ligand and all four CO ligands.

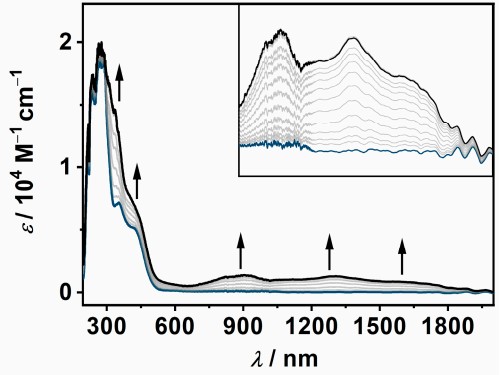 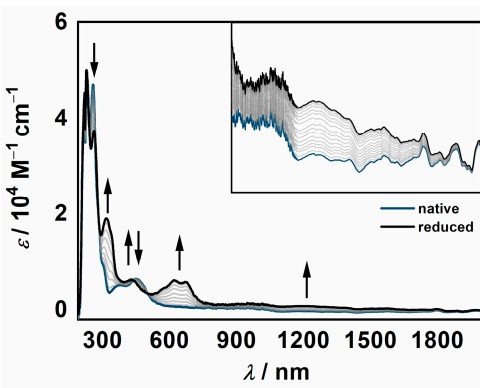

**Figure 5.** Changes in the UV/VIS spectra of [W(**C**–**C**)(CO)$_4$] (**left**, inset: 650–2050 nm) and [W(**C**–**N**)(CO)$_4$] (**right**, inset: 750–2090 nm) in CH$_3$CN/0.1 M Bu$_4$NPF$_6$ during the first reduction with a Au working electrode.

Upon a reduction in [M(**C**–**N**)(CO)$_4$], weak bands are observed in the visible and NIR region (700–2100 nm), which can be assigned to ILCTs and LLCTs from the reduced ligand to the axial CO ligands, the pyridyl-MIC moiety, and the aromatic substituent (see Supplementary Materials S6.40). Additionally, more discrete transitions are observed in the 550–700 nm region, indicating a more localized ligand-centered radical, which is in good agreement with the aforementioned EPR-SEC results. The electronic transitions in this range are best described as a mixture of ILCTs, MLLCTs, and LLCTs.

The partial degradation of [M(**C**–**C**)(CO)$_4$]$^-$ by an EC mechanism is confirmed by the decrease in absorption maxima during electrolysis in the OTTLE cell in the visible and NIR region and further supported by comparing the UV/vis/NIR spectra before and after UV/vis/NIR-SEC (see Supplementary Materials S5.10–S5.30). Only in the case of [Mo(**C**–**C**)(CO)$_4$] could the UV/vis/NIR spectrum of the starting complex be recovered completely. A similar observation was already described by Tory et al., who proposed the recoordination of the CO ligand to the metal center within the experimental setup [39]. In contrast, no degradation within the series of [M(**C**–**N**)(CO)$_4$] is detected, confirming the complete reversibility of the first ligand-centered reduction (see Supplementary Materials S5.40).

Based on our UV/vis/NIR- and IR-SEC measurements, we can conclude that the C–N linkage in [M(**C**–**N**)(CO)$_4$] results in an increased stabilization of the ligand-centered radical, while a reduction in the C–C pyridyl-MIC ligand shows an EC mechanism, leading to CO dissociation (Scheme 3).

An associative mechanism for the CO dissociation is unlikely, as it would generate 21 VE species of the already electron-rich [M(**C**–**C**)(CO)$_4$] complex. Therefore, we propose a dissociative mechanism after the first reduction, leading to 17 VE species [M(**C**–**C**)(CO)$_3$]$^-$. However, the intermediate is coordinatively unsaturated and thus accessible to solvent coordination, generating the complex [M(**C**–**C**)(CH$_3$CN)(CO)$_3$]$^-$.

**Scheme 3.** Simplified mechanism of the one-electron reduction in [M(**C–C**)(CO)$_4$] and [M(**C–N**)(CO)$_4$] (M = Cr, Mo, W) at the Au WE surface (grey: alternative reaction pathways).

Based on the previously described reversibility of [Mo(**C–C**)(CO)$_4$] in UV/vis/NIR-SEC, a stepwise mechanism following the oxidation of the proposed intermediates could lead to the regeneration of the parent complex [39].

Furthermore, the irreversibility of [M(**C–C**)(CO)$_4$] suggests the formation of multiple species after UV/vis/NIR- and IR-SEC (see Supplementary Materials S4.20–S4.40). The newly generated IR bands after IR-SEC at 1888 cm$^{-1}$ and 1776 cm$^{-1}$ (M = W), as well as the IR bands at 1907 cm$^{-1}$ and 1781 cm$^{-1}$ (M = Mo), are in good accordance with the previously reported photo-induced formation of the axial solvent adduct [M(**C–C**)(CH$_3$CN)$_{ax}$(CO)$_3$] after CO dissociation, supporting the proposed EC mechanism [45].

Interestingly, the IR bands of the decomposition products at 1938 cm$^{-1}$ and 1799 cm$^{-1}$ in [W(**C–C**)(CO)$_4$] and 1946 cm$^{-1}$ and 1810 cm$^{-1}$ in [Mo(**C–C**)(CO)$_4$], respectively, are well-described as the *trans*-positioned pyridyl [M(**C–C**)(CH$_3$CN)$_{trans\text{-}N}$(CO)$_3$] and MIC [M(**C–C**)(CH$_3$CN)$_{trans\text{-}C}$(CO)$_3$] solvato complexes, indicating a fluxional reorganization of the CO ligands after electrochemically induced CO dissociation [45].

Inspired by these results, we reinvestigated all the presented complexes by cyclic voltammetry using a Au WE and an electrochemical reduction in CO$_2$ with a GC and Au WE, respectively.

### 2.4. Cyclic Voltammetry with a Au WE and Electrochemical CO$_2$ Reduction

The cyclic voltammograms of [M(**C–C**)(CO)$_4$] and [M(**C–N**)(CO)$_4$] (M = Cr, Mo, W) with a Au WE show the same electrochemical redox processes observed with a GC WE (Figure 6; see Supplementary Materials S2). The second reduction shifts to more cathodic potential, while in the potential range from −1.5 V to ± 0.0 V, only minor changes are observed.

The separation of the first reduction from the second reduction leads to a reversible first reduction in all presented complexes, as indicated by the $\frac{i_c}{i_a}$ current ratio of ≈ 1, the peak-to-peak separation of $\Delta E = 0.07$ V, and the absence of oxidative processes between −1.5 V and ± 0.0 V (Figure 6; see Supplementary Materials S2). The reversibility of the first reduction in the series of [M(**C–C**)(CO)$_4$] (M = Cr, Mo, W) is a direct consequence of the applied scan rate of 100 mV/s, leading to a fast electron transfer processes instead of an EC mechanism, accompanied by CO dissociation, which is observed during electrolysis in the OTTLE cell in the IR- and UV/vis/NIR-SEC measurements.

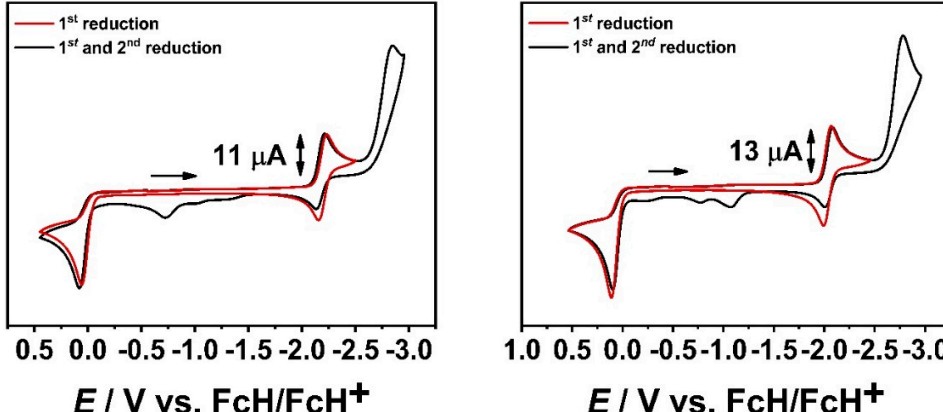

**Figure 6.** Cyclic voltammograms of [W(**C–C**)(CO)$_4$] (**left**) and [W(**C–N**)(CO)$_4$] (**right**) in CH$_3$CN and 0.1 M Bu$_4$NPF$_6$ with a scan rate of 100 mV/s and a Au electrode.

The second reduction appears completely irreversible in the series of [M(**C–C**)(CO)$_4$] and [M(**C–N**)(CO)$_4$] and is further confirmed by the appearance of additional oxidation processes in the range from −1.5 V to ± 0.0 V.

Earlier reports by Hartl and co-workers on [Mo(CO)$_4$(bpy)] showed a reversible first reduction, generating the monoanionic [Mo(CO)$_4$(bpy)]$^-$ species using a Au WE [39]. The second irreversible reduction results in the formation of the coordinatively unsaturated [Mo(CO)$_3$(bpy)]$^{2-}$ complex after CO dissociation. On sweeping back to cathodic potentials, the rapid recoordination of the CO ligand is proposed, as indicated by the near recovery of the first reversible reduction in [Mo(CO)$_4$(bpy)]$^-$.

As judged by the cyclic voltammetry for [M(**C–C**)(CO)$_4$] and [M(**C–N**)(CO)$_4$], no such intermediate could be detected after the second reduction with a Au WE, even at lower scan rates of 25 mV/s (see Supplementary Materials S2). Notably, lowering the scan rate leads to the complete disappearance of the oxidative processes between –1.5 V and ± 0.0 V. Reversible coordination of one of the pyridyl-MIC moieties after the second reduction can, therefore, not be ruled out due to its electron-rich nature [52–55].

In the presence of CO$_2$ under non-protic conditions, the influence of the metal center, the electrode material, and the constitutional isomers reveal their full potential in the electrochemical activation of CO$_2$ (Figure 7; see Supplementary Materials S7).

In the series of [M(**C–C**)(CO)$_4$], only the chromium complex shows a catalytic current with a GC WE after the first catalytic cycle, while no catalytic current is observed for the higher homologs. Instead, an overpotential ($\eta = \sim$280 mV) [56] is observed after the second reduction, which could be a consequence of adduct formation with CO$_2$, leading to the deactivation of the catalysts, as previously reported by Kubiak and co-workers [37].

The second catalytic cycle in [Cr(**C–C**)(CO)$_4$] shows similar reactivity and is described by higher homologs. To verify whether the catalyst is a real homogenous catalyst or deposited on the electrode surface, a rinse test was performed (see Supplementary Materials S58) [57].

As judged by the experimental data, no heterogenous reactivity can be detected, which supports the CO$_2$ adduct formation within the series of [M(**C–C**)(CO)$_4$].

In contrast, in the series of [M(**C–N**)(CO)$_4$], a catalytic current at high potentials $E_p^{cat} > -3.0$ V is detected (see Supplementary Materials S7), which underlines that the fine-tuning of the ligand can have a major impact on catalytic performance. Unfortunately, high applied potentials for the electrocatalytic transformation of CO$_2$ prevented us from further product analysis. Hence, we focused on the influence of the electrode material to shift the onset potential for the electrochemical conversion of CO$_2$ with a Au WE (Figure 7; see Supplementary Materials S7).

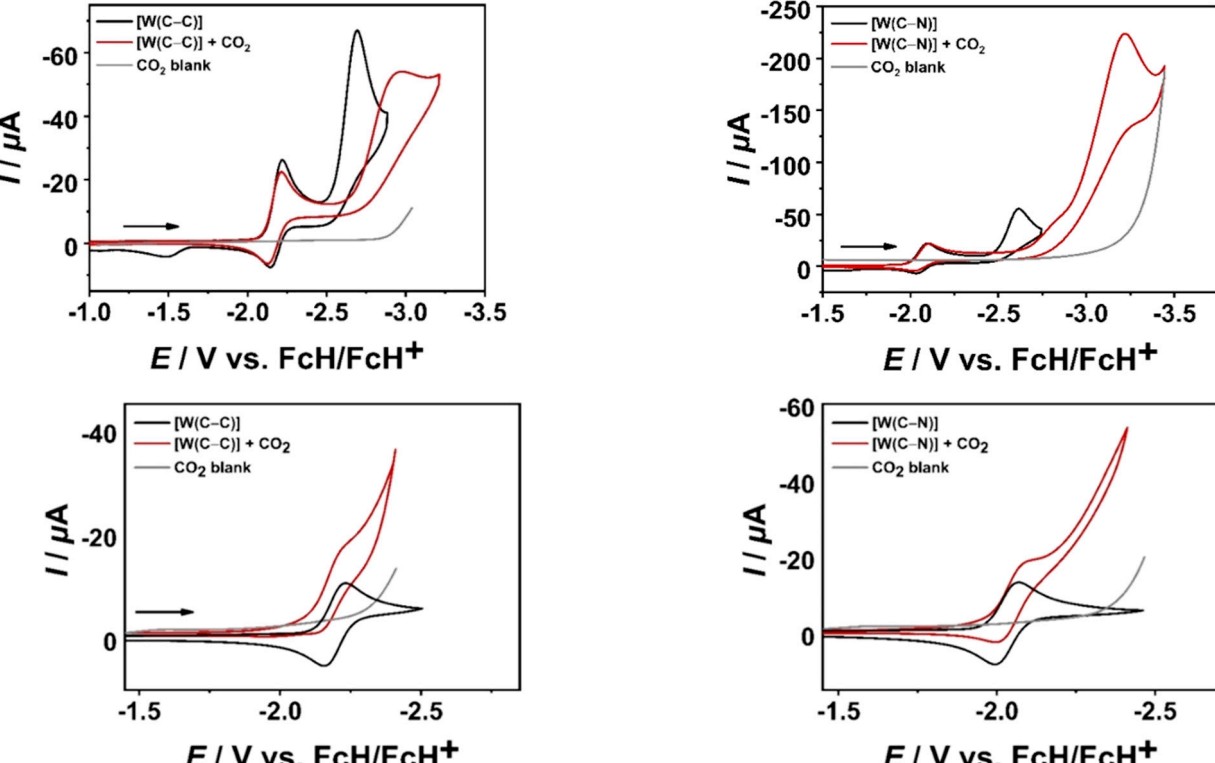

**Figure 7.** Cyclic voltammograms of [W(**C–C**)(CO)$_4$] (**left**) and [W(**C–N**)(CO)$_4$] (**right**) (1 mM, black) and in the presence of CO$_2$ (red) at 100 mV/s in CH$_3$CN/0.1 M Bu$_4$NPF$_6$ with a GC WE (**top**) and a Au WE (**bottom**).

According to our IR-SEC measurements, the first reduction in [M(**C–C**)(CO)$_4$] with a Au WE leads to CO dissociation, creating an open coordination site for binding CO$_2$. However, the weaker $\pi$ acceptor properties of the C–C linked pyridyl-MIC ligand compared to its bpy counterpart shifts the onset potential to higher cathodic potential, preventing us from investigating the catalytic conversion under the experimental conditions, giving access to precatalytic activation (see Supplementary Materials S7).

To our surprise, the electrochemical conversion of CO$_2$ with the greater $\pi$ acceptor ligand in [M(**C–N**)(CO)$_4$] results in a catalytic current close to the potential window of a saturated CO$_2$/CH$_3$CN solution, which is in conflict with our previously described IR-SEC measurements. A plausible explanation could be the formation of traces of [M(**C–N**)(CO)$_3$]$^-$ at the electrode surface, capable of electrocatalytically reducing CO$_2$, as previously described by Cowan and co-workers [36].

Analysis of the results in the electrochemical conversion of CO$_2$ with [M(**C–C**)(CO)$_4$] and [M(**C–N**)(CO)$_4$] using a Au WE show that the onset potential can be shifted drastically, up to +730 mV vs. a GC WE by the right choice of ligand and electrode material, as shown in the case of [W(**C–N**)(CO)$_4$].

## 3. Conclusions

The influence of the constitutional isomers on the redox and the spectroscopic properties of group 6 carbonyl complexes was investigated by cyclic voltammetry, EPR-, IR- and UV/vis/NIR-SEC. According to cyclic voltammetry, the different linkage of the constitutional isomers results in a greater $\sigma$ donor strength of the **C–C** linked pyridyl-MIC ligand and a lower $\pi$ acceptor ability compared to its **C–N** counterpart, which could be further confirmed by IR spectroscopy. The changes in the electronic structure have a tremendous influence on the redox properties of [M(**C–C**)(CO)$_4$] and [M(**C–N**)(CO)$_4$]. Based on our EPR-SEC measurements, the first ligand-centered reduction leads to an increased delocalization

of the electron spin within the **C**–**C** linked isomer. This observation was further supported by UV/vis/NIR-SEC measurements and TD-DFT calculations of the singly-reduced species, indicating an enhanced localization of the charge distribution in $[M(\textbf{C–N})(CO)_4]^-$. Upon reduction, IR-SEC measurements of $[M(\textbf{C–C})(CO)_4]$ show an EC mechanism, leading to CO dissociation using a Au WE, while in the case of $[M(\textbf{C–N})(CO)_4]$, a complete electrochemically reversible one-electron reduction was observed. Additionally, UV/vis/NIR-SEC measurements were performed to confirm the pure reversibility of the first ligand-centered reduction or reversibility following an EC mechanism. In the case of $[Mo(\textbf{C–C})(CO)_4]$, the initial spectra could be fully recovered, indicating reversible binding of CO following an EC mechanism. Based on these results, all presented complexes were further tested for the electrochemical conversion of $CO_2$ using a GC and a Au WE. Performing an electrochemical $CO_2$ reduction with a GC WE indicates that all complexes of the series $[M(\textbf{C–N})(CO)_4]$ are capable of electrochemically converting $CO_2$ at high potentials, while the $[M(\textbf{C–C})(CO)_4]$ complexes tend to generate $CO_2$ adducts after the second reduction. A change in electrode material leads to a shift of the onset potential of about +730 mV. However, the catalytic performance close to the potential window of the $CO_2$-saturated 0.1 M $CH_3CN/Bu_4NPF_6$ precluded further analysis of the product formation. Qualitatively, all presented complexes are capable of activating $CO_2$ by changing the working electrode from GC to Au. In this study, we were able to demonstrate that minor changes in the ligand framework, metal center, and experimental setup can have a tremendous influence on the electrochemical, spectroelectrochemical, and electrocatalytic performance in such systems.

## 4. Experimental Section

The synthesis of the complexes $[M(\textbf{C–C})(CO)_4]$ and $[M(\textbf{C–N})(CO)_4]$ (M = Cr, Mo, W) was performed according to the previously reported literature procedures [41–43,45].

### 4.1. General Procedures, Materials, and Instrumentation

**Caution!** Compounds containing azides are potentially explosive. Although we never experienced any problems during synthesis or analysis, all compounds should be synthesized in small quantities and handled with great care!

Unless otherwise noted, all reactions were carried out using standard Schlenk-line techniques under an inert atmosphere of argon (Linde Argon 4.8, purity 99.998%) or in a glovebox (Glovebox Systemtechnik, GS095218).

Commercially available chemicals were used without further purification. The solvents used for metal complex synthesis and catalysis were available from MBRAUN MB-SPS-800 solvent System and degassed by standard techniques prior to use. The identity and purity of the compounds were established via $^1$H and $^{13}$C NMR spectroscopy, elemental analysis, and mass spectrometry.

Solvents for cyclic voltammetry and UV/vis- and EPR-spectroelectrochemical measurements were dried and distilled under argon and degassed by common techniques prior to use. Column chromatography was performed over silica 60 M (0.04–0.063 mm).

$^1$H and $^{13}$C{$^1$H} NMR spectra were recorded on a Bruker Advance 400 spectrometer at 19–22 °C. Chemical shifts are reported in ppm referenced to the residual solvent peaks [58].

The following abbreviations are used to represent the multiplicity of the signals: s (singlet), d (doublet), t (triplet), q (quartet), p (pentet), and sept (septet).

Mass spectrometry was performed on an Agilent 6210 ESI-TOF.

Elemental analyses were performed with an Elementar Micro Cube elemental analyzer.

The light-induced syntheses were performed with a LOT-QuantumDesign Arc Lamp (150 W, Xe OF).

### 4.2. Electrochemistry

Cyclic voltammograms were recorded with a PalmSens4 potentiostat or PAR VersaStat (Ametek), respectively, with a conventional three-electrode configuration consisting of a glassy carbon working electrode or gold working electrode, a platinum auxiliary electrode,

and a coiled silver wire as a pseudoreference electrode. The ferrocene/ferrocenium couple was used as an internal reference. All measurements were performed at room temperature at a scan rate between 25 and 1000 mVs$^{-1}$. The experiments were carried out in absolute Acetonitrile containing 0.1 M Bu$_4$NPF$_6$ (Sigma Aldrich, $\geq$99.0%, electrochemical grade) as the supporting electrolyte.

### 4.3. Spectroelectrochemistry

UV/vis spectra were recorded with an Avantes spectrometer consisting of a light source (AvaLight-DH-S-Bal), a UV/vis detector (AvaSpec-ULS2048), and an NIR detector (AvaSpeC-NIR256-TEC). IR spectra were recorded with a BRUKER Vertex 70 FT-IR or Nicolet 6700 FT-IR spectrometer, respectively. UV/vis-spectroelectrochemical measurements were carried out in an optically transparent thin-layer electrochemical (OTTLE) [59,60] cell (CaF$_2$ windows) with a gold-mesh working electrode, a platinum-mesh counter electrode, and a silver-foil pseudoreference. EPR spectra at the X-band frequency (ca. 9.5 GHz) were obtained with a Magnettech MS-5000 benchtop EPR spectrometer equipped with a rectangular TE 102 cavity and a TC HO4 temperature controller. The measurements were carried out in synthetic quartz glass tubes. For EPR spectroelectrochemistry, a three-electrode setup was employed using two Teflon-coated platinum wires (0.005 in. bare and 0.008 in. coated) as the working and counter electrodes and a Teflon-coated silver wire (0.005 in. bare and 0.007 in coated) as the pseudoreference electrode. The experiments were carried out in absolute Acetonitrile containing 0.1 M Bu$_4$NPF$_6$ as the supporting electrolyte. The same solvents used for CV measurements were used for each compound.

### 4.4. Calculations

The program package ORCA 4.1 was used for all DFT calculations [61]. Starting from the molecular structure obtained from X-ray diffraction, geometry optimizations were carried out using the B3LYP [62,63] function, and no symmetry restrictions were imposed during the optimization. For tungsten, relativistic effects in zero-order regular approximation (ZORA) were included [64]. All calculations were performed with an empirical Van der Waals correction (D3) [65–68]. The restricted and unrestricted DFT methods were employed for closed and open shell molecules, respectively, unless stated otherwise. Convergence criteria were set to the default for geometry optimization (OPT) and tight for SCF calculations (TIGHTSCF). Triple-$\zeta$ valence basis sets (def2-TZVP) [69] were employed for all atoms. Calculations were performed using a resolution of the identity approximation [70–76] with matching auxiliary basis sets [77,78] for geometry optimizations and numerical frequency calculations, and a RIJCOSX (combination of the resolution of the identity and chain of spheres algorithms) approximation was used for single-point calculations using the B3LYP function. Low-lying excitation energies were calculated with time-dependent DFT (TD-DFT). Solvent effects were taken into account with the conductor-like polarizable continuum model, CPCM [79]. Spin densities were calculated according to the Mulliken population analysis [80]. The absence of imaginary frequency, spin densities, molecular orbitals, and difference densities were visualized with a modified Chemcraft 1.8 program [81,82]. All molecular orbitals are illustrated with an iso value of 0.052. All calculated TD-DFT spectra are Gaussian-broadened with a bandwidth of 25 at half height unless otherwise noted.

**Supplementary Materials:** The following supporting information can be downloaded at https://www.mdpi.com/article/10.3390/inorganics12020046/s1.

**Author Contributions:** Conceptualization, T.B. and B.S.; methodology, T.B.; software, T.B.; validation, T.B. and B.S.; formal analysis, T.B. and B.S.; investigation, T.B.; resources, B.S.; data curation, T.B.; writing—original draft preparation, T.B.; writing—review and editing, B.S.; visualization, T.B.; supervision, B.S.; project administration, B.S.; funding acquisition, B.S. All authors have read and agreed to the published version of the manuscript.

**Funding:** We thank the state of Baden-Württemberg through bwHPC and the German Research Foundation (DFG) through grant No. INST 40/575-1 FUGG (JUSTUS 2 cluster) for their support.

**Data Availability Statement:** The data that support the findings of this study are available in the Supporting Material in this article.

**Acknowledgments:** We thank the state of Baden-Württemberg through bwHPC and the German Research Foundation (DFG) through grant No. INST 40/575-1 FUGG (JUSTUS 2 cluster) for their support.

**Conflicts of Interest:** There are no conflicts of interest to declare.

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
