# Peer review of "Investigations of the Influence of Two Pyridyl-Mesoionic Carbene Constitutional Isomers on the Electrochemical and Spectroelectrochemical Properties of Group 6 Metal Carbonyl Complexes"

_inorganics, doi:10.3390/inorganics12020046_

Round 1

Reviewer 1 Report

Comments and Suggestions for Authors

In the present work the investigators are studying the electronic effect of substituting for the purpose of preparing metal complexes of group 6 metals which are capable of electrocatalytically reducing carbon dioxide. The interaction of the catalyst with the electrode was investigated by vibrational sum frequency generation spectroscopy (VSFG). EPR studies are also included to probe ligand coupling.

The work is nicely carried out and supports the conclusion of the manuscript. On page 4 lines 118-27, the authors give a good discussion on the CV and adequately explain the irreversible waves which are likely due to CO loss. In the EPR studies, the authors do an excellent job of supporting the experimental results with simulations and other theoretical studies. The interpretation of EPR spectra appear to be accurate. The interpretation of the IR results appears accurate. the associative mechanism proposal on page 11 (line 282) is a reasonable suggestion and is consistent with what would be expected based on the 18 electron rule. Overall there is a good progression in experimental strategies and limitations in the study are pointed out. The conclusion is a good summary of the experimental results.

Items that need attention:

-I believe there is some formatting issues with the C-C and C-N ligand representation. Also, page 9 line 240 (300-550 nm).

-Please define FcH in the electrochemistry section. FcH is used for ferrocene but in the experimental (Cp*)2Fe is mentioned.

-On page 7, line 170, there is an extra space after [W(C-N)(CO)4].

-Page 8, line 205: The isosbestic point does not confirm a clean conversion of one species to another but is consistent with this result.

-Page 13, line 364: This should read "preventing us from investigating the catalytic..."

Author Response

-I believe there is some formatting issues with the C-C and C-N ligand
representation. Also, page 9 line 240 (300-550 nm).

These formatting issues have been corrected.

-Please define FcH in the electrochemistry section. FcH is used for
ferrocene but in the experimental (Cp*)2Fe is mentioned.

FcH has been defined in Table 1. Decamethylferrocene was a mistake and has been corrected.

-On page 7, line 170, there is an extra space after [W(C-N)(CO)4].

This has been corrected.

-Page 8, line 205: The isosbestic point does not confirm a clean
conversion of one species to another but is consistent with this result.

This sentence has been reformulated.

-Page 13, line 364: This should read "preventing us from investigating
the catalytic..."

This sentence has been corrected.

Reviewer 2 Report

Comments and Suggestions for Authors

Good paper.

Page 2, line 79: “L: C C = 79 pyridyl-4-triazolylidene [41] and C N = pyridyl-1-triazolylidene” instead of C-C and C-N there is a space between C C and C N. This appears everyplace where there is a mention of this ligand system.

Page 10, line 248: similar situation, “600 1200 nm” should be 600-1200 nm. Line 258 has same appearance for 550-700 nm.

Page 12, line 326: “coordinatively unsaturated 326 [Mo(CO)3(bpy)]2 ”, Rephrase without superscript 2.

Page 16, “Supplementary Materials: The following supporting information can be downloaded at: www.mdpi.com/xxx/s1, Figure S1: title; Table S1: title; Video S1: title”; needs accurate representation, no video was detected for review.

Page 16, “Funding: Please add: “This research received no external funding” or “This research was funded by 489 NAME OF FUNDER, grant number XXX” and “The APC was funded by XXX”. Check carefully that 490 the details given are accurate and use the standard spelling of funding agency names at 491 https://search.crossref.org/funding. Any errors may affect your future funding”; needs specific information or needs to be omitted.

Author Response

Page 2, line 79: “L: C C = 79 pyridyl-4-triazolylidene [41] and C N = pyridyl-1-triazolylidene” instead of C-C and C-N there is a space between C C and C N. This appears everyplace where there is a mention of this ligand system.

-In the word document that we are submitting now, all the “-“ can be seen.

Page 10, line 248: similar situation, “600 1200 nm” should be 600-1200 nm. Line 258 has same appearance for 550-700 nm.

-The “-“ can also be seen here.

Page 12, line 326: “coordinatively unsaturated 326 [Mo(CO)3(bpy)]2 ”, Rephrase without superscript 2.

-This has been rephrased.

Page 16, “Supplementary Materials: The following supporting information can be downloaded at: www.mdpi.com/xxx/s1, Figure S1: title; Table S1: title; Video S1: title”; needs accurate representation, no video was detected for review.

-This sentence has been modified.

Page 16, “Funding: Please add: “This research received no external funding” or “This research was funded by 489 NAME OF FUNDER, grant number XXX” and “The APC was funded by XXX”. Check carefully that 490 the details given are accurate and use the standard spelling of funding agency names at 491 https://search.crossref.org/funding. Any errors may affect your future funding”; needs specific information or needs to be omitted.

  • Specific information has been provided.